# Mucosal infection rewires TNFα signaling dynamics to skew susceptibility to recurrence

Lu Yu[1†], Valerie P O'Brien[1†], Jonathan Livny[2], Denise Dorsey[1], Nirmalya Bandyopadhyay[2], Marco Colonna[3], Michael G Caparon[1], Elisha DO Roberson[4,5], Scott J Hultgren[1]*, Thomas J Hannan[1,3]*

[1]Department of Molecular Microbiology and Center for Women's Infectious Disease Research, Washington University School of Medicine, St Louis, United States; [2]The Broad Institute of Massachusetts Institute of Technology and Harvard, Cambridge, United States; [3]Department of Pathology and Immunology, Washington University School of Medicine, St Louis, United States; [4]Department of Medicine, Division of Rheumatology, Washington University School of Medicine, St Louis, United States; [5]Department of Genetics, Washington University School of Medicine, St Louis, United States

**Abstract** A mucosal infectious disease episode can render the host either more or less susceptible to recurrent infection, but the specific mechanisms that tip the balance remain unclear. We investigated this question in a mouse model of recurrent urinary tract infection and found that a prior bladder infection resulted in an earlier onset of tumor necrosis factor-alpha (TNFα)-mediated bladder inflammation upon subsequent bacterial challenge, relative to age-matched naive mice. However, the duration of TNFα signaling activation differed according to whether the first infection was chronic (Sensitized) or self-limiting (Resolved). TNFα depletion studies revealed that transient early-phase TNFα signaling in Resolved mice promoted clearance of bladder-colonizing bacteria via rapid recruitment of neutrophils and subsequent exfoliation of infected bladder cells. In contrast, sustained TNFα signaling in Sensitized mice prolonged damaging inflammation, worsening infection. This work reveals how TNFα signaling dynamics can be rewired by a prior infection to shape diverse susceptibilities to future mucosal infections.
DOI: https://doi.org/10.7554/eLife.46677.001

*For correspondence:
hultgren@wustl.edu (SJH);
thannan@wustl.edu (TJH)

†These authors contributed equally to this work

Competing interests: The authors declare that no competing interests exist.

## Introduction

Mucosal bacterial infections are very common, accounting for over 42 million outpatient visits and a majority of the 270 million outpatient antibiotic prescriptions in the United States annually (*Armstrong and Pinner, 1999*; *May et al., 2014*; *Outpatient antibiotic prescriptions, 2011*). Of these, respiratory and urinary tract infections (UTI) can be highly recurrent, the latter leading to over $2 billion in direct and indirect costs annually in the United States (*Foxman, 2010*). Over 80% of community-acquired UTI are caused by uropathogenic *Escherichia coli* (UPEC) (*Gupta and Bhadelia, 2014*; *Ronald, 2003*), and the vast majority of these infections involve the lower urinary tract and specifically the urinary bladder, causing cystitis (bladder infection). Women are disproportionately affected: over 60% of women will experience at least one UTI during their lifetime (*Foxman et al., 2000a*), and of these women, 20–30% will experience a recurrence (rUTI) within 6 months (*Foxman et al., 2000b*). Some individuals experience recurrent UTI to such a high degree that they resort to taking daily suppressive antibiotic therapy to prevent further recurrence (*Ikäheimo et al., 1996*). In sexually active young women, one of the single biggest independent risk factors for

developing an acute UTI is a history of two or more UTIs (*Hooton et al., 1996*). Furthermore, placebo studies of UTI patients show that the natural course of infection and symptoms can vary greatly: for some women, infections resolve spontaneously within a few days, while others may develop persistent bladder infections lasting for weeks (*Ferry et al., 2004*; *Christiaens et al., 2002*; *Mabeck, 1972*). Taken together, these data suggest that disease history may impact the nature of the future interaction of UPEC with the urinary tract mucosa of patients in a way that alters susceptibility to symptomatic rUTI.

In C3H/HeN mice, a dichotomy of self-resolving vs. chronic infection, similar to the natural course of UTI in women, occurs after experimental UPEC infection in an infectious dose-dependent manner (*Hannan et al., 2010*). Strikingly, the outcome of the initial infection polarizes future susceptibility to rUTI: mice that spontaneously resolve the initial infection are highly resistant to rUTI upon challenge, whereas those that develop an initial chronic infection (persistent high-titer bladder bacterial colonization and bladder mucosal inflammation) lasting two weeks or longer prior to antibiotic therapy are highly sensitive to rUTI (*O'Brien et al., 2015*). These studies revealed an early immune checkpoint that determines whether the host spontaneously resolves the infection or goes on to develop chronic cystitis (*Hannan et al., 2010*). Specifically, at 24 hours post-inoculation (hpi), elevated serum cytokines (IL-5, IL-6, G-CSF, CXCL1), severe bladder inflammation with mucosal wounding, and pyuria (neutrophils in urine) are the hallmarks of checkpoint activation, which indicates severe acute UTI that predisposes to chronic infection. In mice that trigger the checkpoint, robust cyclooxygenase-2 (COX-2) expression by bladder epithelial (urothelial) cells at 24 hpi promotes neutrophil transmigration through the urothelium, causing mucosal damage and ultimately the development of chronic cystitis (*Hannan et al., 2014*). If chronic cystitis is permitted to go on for two or more weeks prior to antibiotic therapy, affected mice then become highly susceptible to developing severe rUTI upon subsequent challenge in a COX-2 dependent manner (*Hannan et al., 2010*; *Hannan et al., 2014*; *O'Brien et al., 2016*). C57BL/6J mice, though inherently resistant to chronic cystitis upon a single bladder inoculation of UPEC, can be induced to develop chronic cystitis by repeated bacterial inoculation (*Hannan et al., 2010*; *Schwartz et al., 2015*), and those with a history of chronic infection become susceptible to severe rUTI upon challenge, similar to C3H/HeN mice, demonstrating that acquired factors related to a prior chronic infection can overcome genetic resistance (*O'Brien et al., 2016*). Translating these findings to humans, we found that serum cytokine biomarkers associated with more robust granulocytic responses, similar to those associated with the development of chronic cystitis in naive mice, were also predictive of susceptibility to rUTI in women (*Hannan et al., 2014*).

The diverging susceptibilities of mice to rUTI as a consequence of the outcome of a prior infection were found to be due in part to bladder remodeling that alters the pathophysiology of acute cystitis upon the second infection (*Hannan et al., 2014*; *O'Brien et al., 2016*). However, the molecular mechanisms by which this bladder remodeling impacted susceptibility to rUTI were unclear. Here, we leveraged the power of bladder transcriptomics and the C3H/HeN mouse model of rUTI to interrogate the molecular basis for the impact of UTI history on host susceptibility to rUTI. We discovered that tumor necrosis factor-alpha (TNFα) signaling was a hallmark of bladder inflammation during acute UTI in mice. However, the temporal nature of the TNFα response differed according to UTI history and impacted the outcome of recurrent infection. Antibody-mediated depletion experiments showed that the polarizing rUTI susceptibility phenotypes (i.e. mice with a history of chronic UTI were susceptible to rUTI, whereas mice with a history of self-resolving UTI were resistant to rUTI) were each mediated by TNFα signaling, but with divergent dynamics and effects. Thus, these findings demonstrate the central role of TNFα in directing acute bladder inflammation and provide mechanistic insight into how UTI history affects host susceptibility to recurrent infection by modulating TNFα signaling dynamics.

## Results

### Infection history alters acute bladder inflammation kinetics during recurrent UTI

To assess the contribution of host infection history to rUTI susceptibility, we utilized a previously described recurrent cystitis model in adult (16 weeks old) C3H/HeN mice (*Figure 1A*). Relative to

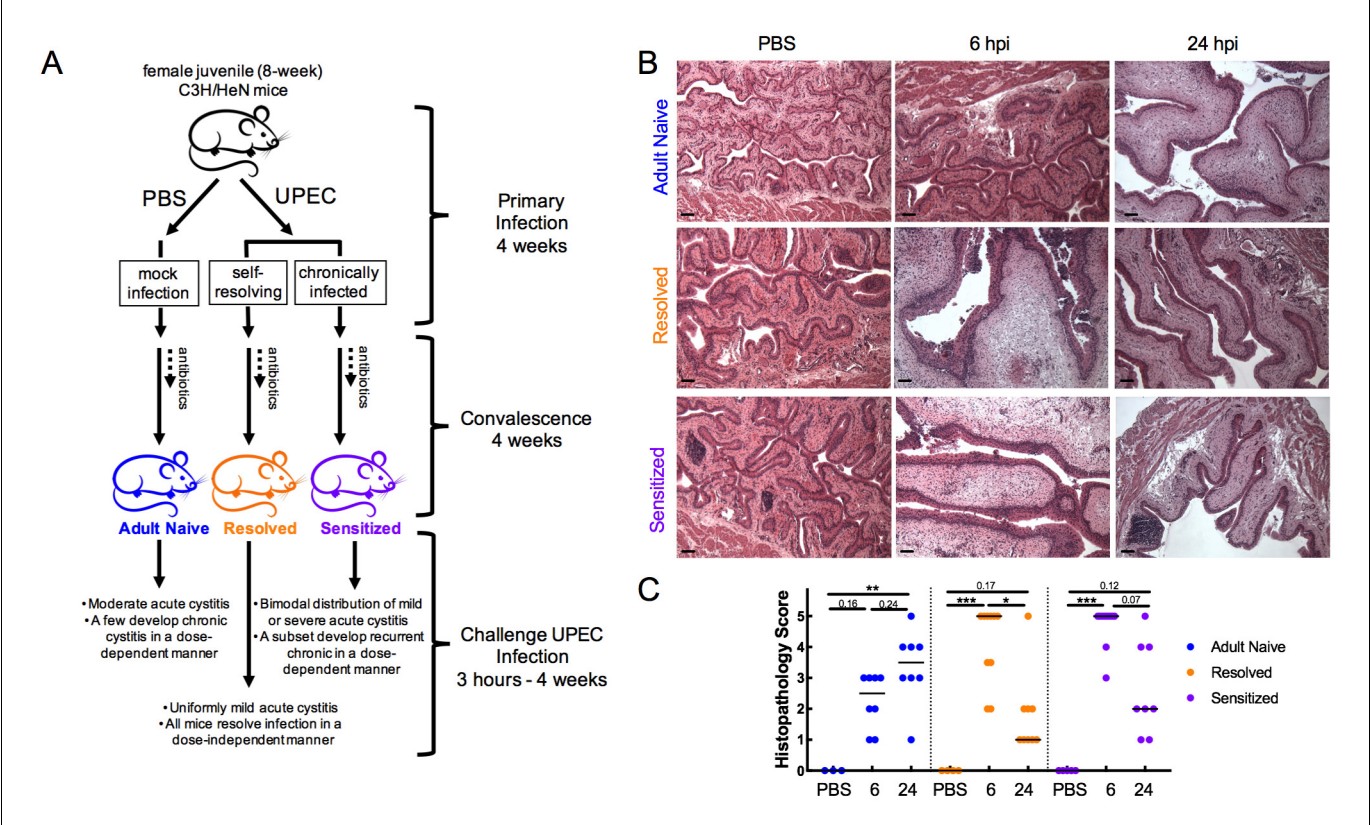

**Figure 1.** Infection history alters acute bladder inflammation kinetics during recurrent UTI. (**A**) An illustration of C3H/HeN recurrent cystitis model. (**B**) and (**C**) C3H/HeN Resolved and Sensitized mice, and Adult Naive mice as a control, were infected with $10^8$ cfu UTI89 or PBS and sacrificed at denoted time points. (**B**) Hematoxylin and eosin-stained bladder sections were assessed in a blinded fashion. Representative images are shown; scale bars = 50 μm. (**C**) Histopathology scores were determined as described in the Materials and methods and the degree of inflammation was assessed based on levels of immune cell infiltration, edema, and urothelial integrity. N = 2 independent experiments were conducted. Data points represent values for each individual mouse, bars indicate median values. Actual *P* values are indicated on the graphs if >0.05, or are represented by the following symbols: *p<0.05, **p<0.01, ***p<0.001, Kruskal-Wallis test with Dunn's correction for multiple comparisons.

DOI: https://doi.org/10.7554/eLife.46677.002

The following source data is available for figure 1:

**Source data 1.** Infection history alters acute bladder inflammation kinetics during recurrent UTI.

DOI: https://doi.org/10.7554/eLife.46677.003

age-matched Adult Naive mice (no history of infection), C3H/HeN mice that spontaneously resolve an initial acute bladder infection are referred to herein as 'Resolved.' In contrast, mice with a history of chronic cystitis lasting two weeks or more prior to antibiotic therapy (which sterilizes the bladder) are referred to herein as 'Sensitized.' In response to challenge infection, the majority of Sensitized mice develop severe recurrent cystitis that often becomes chronic, whereas cystitis in Adult Naive mice is more moderate and few mice develop chronic cystitis. In contrast, Resolved mice are highly resistant to severe recurrent cystitis during the challenge infection and none develop chronic infections (*Hannan et al., 2010*; *O'Brien et al., 2016*).

We performed challenge UPEC infections in Resolved and Sensitized mice four weeks after the initiation of a ten-day course of antibiotics to clear the initial infection. Previous studies have found that this 'convalescence' period is sufficient for healing of the bladder mucosa and a return to a non-inflamed, though remodeled, state. In Sensitized mice this remodeling is evidenced by a hyperplastic urothelium that has a defect in terminal differentiation and the presence of persistent lymphoid follicles, which are not necessary for sensitization but often appear with extended (longer than two weeks) duration of initial infection (*Hannan et al., 2010*; *O'Brien et al., 2016*). Age-matched Adult Naive mice that were initially mock-infected with PBS in parallel served as a control (*Figure 1A*).

Histopathological analysis of the urinary bladder after challenge with $10^8$ colony-forming units (cfu) of the UPEC strain UTI89 revealed significant differences in the severity and kinetics of bladder inflammation in the previously infected mouse groups relative to Adult Naive mice (*Figure 1B–C*). Whereas the majority of Resolved and Sensitized mice had severe bladder inflammation (median bladder histopathology scores of 5 for each group) at 6 hpi, none of the Adult Naive mice had severe bladder inflammation (median histopathology score of 2.5). By 24 hpi, the severity of bladder inflammation had waned significantly in Resolved mice and trended lower in Sensitized mice (median bladder histopathology scores of 1 and 2, respectively), but had increased in Adult Naive mice, though not to a severe level in most individuals, with a median bladder histopathology score of 3.5.

## Infection history alters the course of bladder pathophysiology during recurrent UTI

Consistent with previous work using a $10^7$ cfu challenge (*O'Brien et al., 2016*), we observed significantly lower titers at 6 hpi in mice with a history of UTI relative to Adult Naive mice (*Figure 2A*), indicating that the enhanced bladder inflammation at 6 hpi in previously infected mice was not a consequence of higher bacterial burdens. At 24 hpi, a bimodal distribution of bladder bacterial burdens was observed in mice with a previous history of UTI, with 11 of 14 Sensitized mice experiencing a bloom of UPEC to levels between $10^5$ and $10^8$ cfu/bladder. These titers are similar to or higher than those seen in Adult Naive mice, with an increase in overall median bacterial burden from $3.2 \times 10^4$ at 6 hpi to $8.6 \times 10^5$ cfu/bladder at 24 hpi. In contrast, the median bacterial burden in Resolved mice decreased from $1.7 \times 10^5$ at 6 hpi to $1.6 \times 10^4$ cfu/bladder at 24 hpi, though 6 of 17 mice had high bacterial burdens above $10^5$ cfu/bladder at 24 hpi (*Figure 2A*). However, this subset of Resolved mice with higher bladder titers at 24 hpi did not reflect a propensity for chronic infection, as only one of 25 Resolved mice infected with $10^8$ cfu UTI89 developed chronic infection lasting four weeks, levels similar to those seen in 'resistant' C57BL/6J mice (*Hannan et al., 2010*). The pattern of bladder edema as indicated by the weights of these bladders mirrored the patterns of bladder inflammation observed by histopathology (*Figure 2B*). Most notably, previously infected mice, whether Resolved or Sensitized, had significantly increased bladder weights at 6 hpi (relative to mock-infected mice with the same infection history) that by 24 hpi had decreased significantly in Resolved mice but remained elevated in Sensitized mice. Bladder weights in Adult Naive mice were not significantly increased until 24 hpi. Bladder levels of the pro-inflammatory cytokines IL-6, CXCL1, and CCL2 followed similar patterns, with sharply increased levels at 6 hpi in previously infected mice relative to Adult Naive mice and waning of these levels by 24 hpi in Resolved, but not in Sensitized, mice. In Adult Naive mice, a gradual increase of these cytokines was observed from 6 hpi to 24 hpi (*Figure 2C*). Taken together, these data demonstrate that a prior UTI in C3H/HeN mice confers the ability to mount a robust early phase (first 6 hpi) inflammatory response to UPEC infection not seen in Adult Naive mice, and that this inflammation is more likely to be sustained in Sensitized mice.

## Isogenic mice with different infection histories show distinct bladder transcriptomic profile kinetics

To explore the transcriptomic signatures driving the different patterns of inflammatory responses in C3H/HeN mice with different UTI histories, we performed multiplexed, paired-end RNA-seq experiments on rRNA-depleted RNA isolated from whole bladders. Adult Naive, Sensitized, and Resolved mice were infected with $10^8$ cfu UTI89 or mock-infected with PBS. Differential gene expression was then determined by comparing samples obtained from Adult Naive, Sensitized and Resolved mice at 3.5, 6 and 24 hpi to bladders with the same infection history but mock-infected with PBS for 3.5 hr. Gene fold changes in UPEC-infected mice were calculated relative to mock-infected counterparts using normalized reads. The transcriptomic profiles displayed patterns similar to those seen in the inflammatory profiles (*Figure 3A*). In Adult Naive mice, the number of differentially expressed genes (DEGs) increased with time, starting low (291 genes at 3.5 hpi), increasing somewhat by 6 hpi (5309 genes), and peaking at 24 hpi (9249 genes). However, in both Resolved and Sensitized mice, the number of DEGs was relatively low at 3.5 hpi (127 and 944 genes, respectively), but peaked at 6 hpi (10872 and 10133 genes, respectively), and decreased somewhat by 24 hpi (7793 and 8252 genes, respectively; *Figure 3B*). To obtain a global view of the interplay among infection history, infection status, and gene expression, principal component analyses (PCA) were performed. The PCA plot of

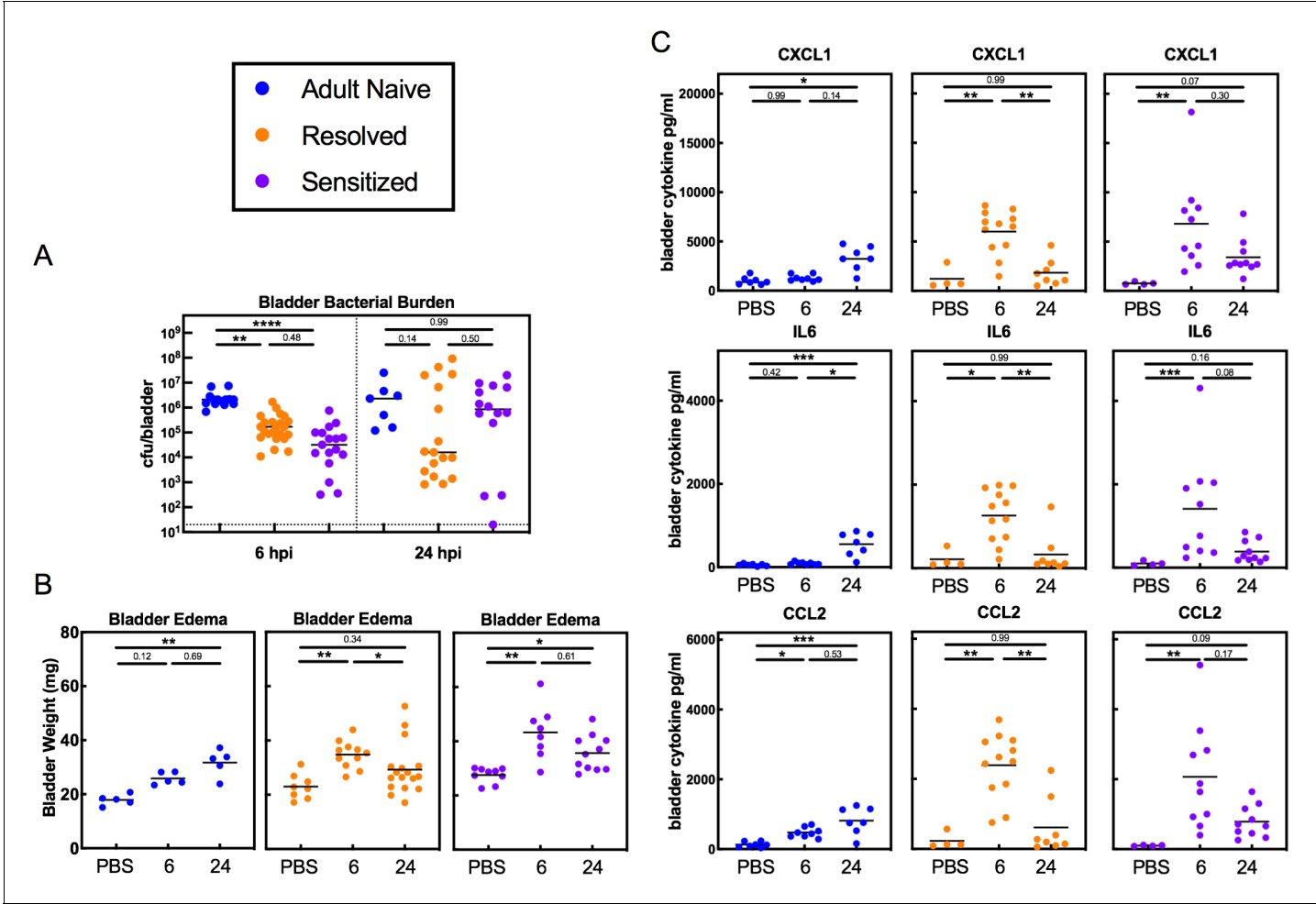

**Figure 2.** Infection history alters the course of bladder pathophysiology during recurrent UTI. C3H/HeN Resolved and Sensitized mice, and Adult Naive mice as a control, were infected with $10^8$ cfu UTI89 or PBS and sacrificed at 6 or 24 hpi. (**A**) Bladder bacterial burdens (cfu/bladder) and (**B**) bladder edema were assessed in N = 4 independent experiments. Bladder edema was assessed by measuring the wet tissue weight of bladders immediately after sacrifice. (**C**) Levels of the cytokines CXCL1, IL-6, and CCL2 were assessed by ELISA of bladder homogenate supernatants from samples shown in panel C, which were collected from N = 3 independent experiments and assayed simultaneously in duplicate. Data points represent values for each individual mouse (the ELISA values are the average of two technical replicates), bars indicate median values, negative results are plotted at the limit of detection (dotted line). Actual P values are indicated on the graphs if >0.05, or are represented by the following symbols: *p<0.05, **p<0.01, ***p<0.001, ****p<0.0001, Kruskal-Wallis test with Dunn's correction for multiple comparisons.
DOI: https://doi.org/10.7554/eLife.46677.004

The following source data is available for figure 2:

**Source data 1.** Infection history alters the course of bladder pathophysiology during recurrent UTI.
DOI: https://doi.org/10.7554/eLife.46677.005

the mock-infected mice demonstrated that Resolved mice were similar to Adult Naive mice (*Figure 3C*). However, Sensitized mice clustered separately, likely due in part to the presence of lymphoid follicles, similar to a previous study (*O'Brien et al., 2016*). The number of genes that were differentially expressed between the mock-infected Sensitized vs. Resolved groups (1019 genes) was similar to the number found in a previous study (992 genes) (*O'Brien et al., 2016*), with 357 of the differentially expressed genes and 98 of the 139 enriched pathways being the same between studies (*Figure 3—figure supplement 1*). Combining all the experimental groups in one PCA shows that the mice infected for 6 or 24 hr clustered separately from mice infected for 3.5 hr or mock-infected on the PC1 axis, regardless of infection history. This clustering is likely driven by infiltration of immune cells into the bladder tissue between 3.5 and 6 hpi in all mice. In contrast, Sensitized mice

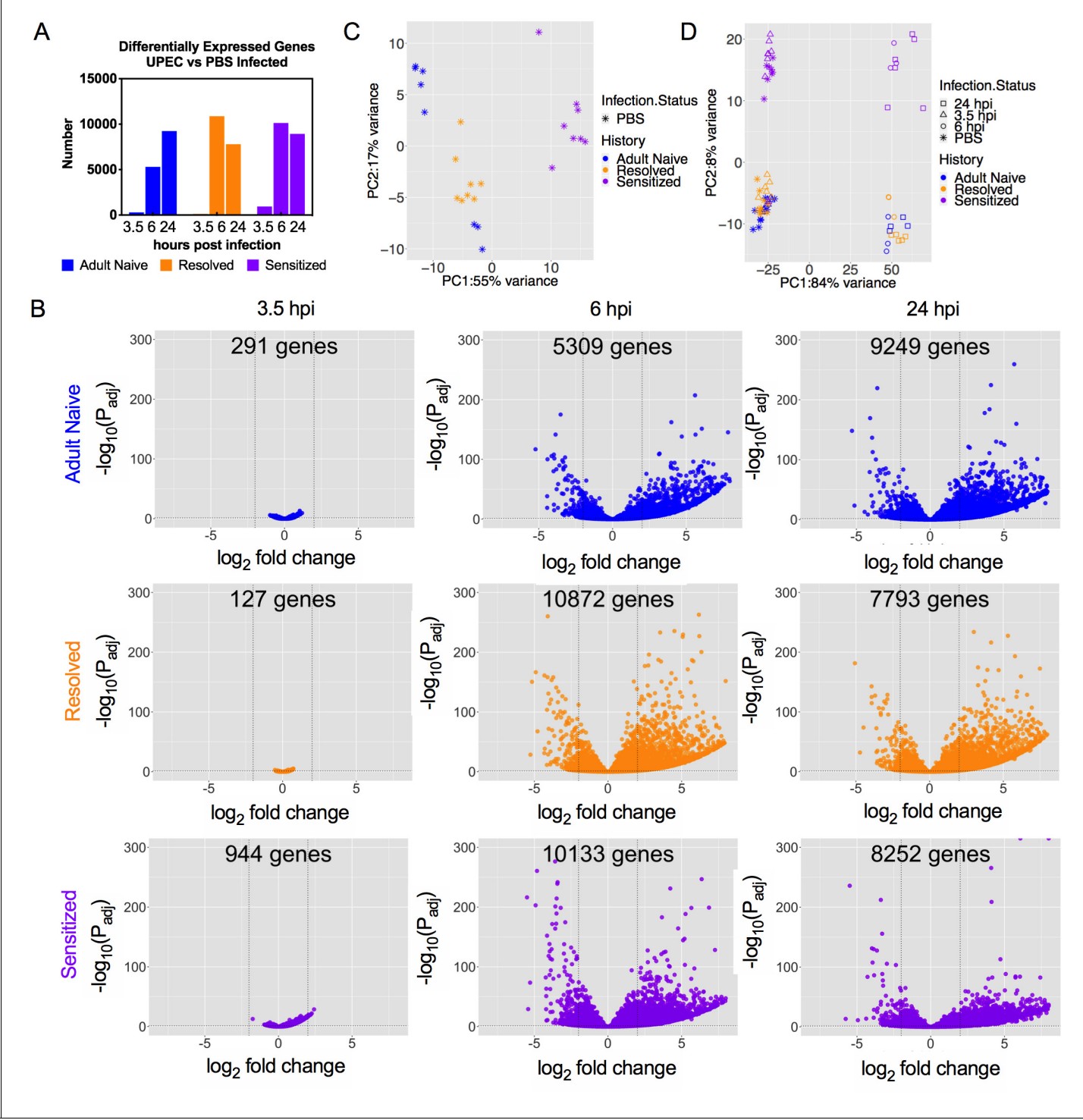

**Figure 3.** Isogenic mice with different infection histories showed distinct bladder transcriptomic profile kinetics. C3H/HeN Resolved and Sensitized mice, and Adult Naive mice as a control, were infected with $10^8$ cfu UTI89 or mock-infected with PBS. RNA was isolated from whole bladders obtained from mice at 3.5, 6, or 24 hpi (eight mice per group for mock-infected and 3.5 hpi; three mice per group at 6 hpi; at 24 hpi, 4 Adult Naive, 5 Resolved, 6 Sensitized) in N = 4 independent infection experiments and grouped into N = 2 independent sequencing experiments. All mock-infected bladders were harvested at 3.5 hpi for baseline controls. (**A**) The number of significantly differentially expressed genes at each time point compared to mock-infected mice with the same infection history. (**B**) Volcano plot of statistically significantly differentially expressed genes (DEGs) of Adult Naive, Resolved, and Sensitized mice (infected vs. mock-infected); the number of DEGs is denoted on each graph. (**C**) Principal component analysis (PCA) of gene expression in mock-infected mice. See also *Figure 3—figure supplement 1*. (**D**) PCA of gene expression in all mouse groups at all time points,

*Figure 3 continued on next page*

*Figure 3 continued*

with shapes indicating different time points post-infection. Each dot represents the transcriptomic profile of a mouse. The PC1 and PC2 axis labels in C) and D) represent principal components 1 and 2, respectively, followed by the percentage of variance they account for.

DOI: https://doi.org/10.7554/eLife.46677.006

The following source data and figure supplement are available for figure 3:

**Source data 1.** Isogenic mice with different infection histories showed distinct bladder transcriptomic profile kinetics.

DOI: https://doi.org/10.7554/eLife.46677.008

**Figure supplement 1.** Comparison of current RNA-seq analysis to prior similar work.

DOI: https://doi.org/10.7554/eLife.46677.007

clustered separately from Resolved and Adult Naive mice on the PC2 axis, irrespective of time point (*Figure 3D*).

## TNFα receptor pathway activation dynamics correlate with host susceptibility to recurrent cystitis

Pathway analysis of the biological processes represented by the DEGs showed that in all mouse groups (Adult Naive, Resolved, and Sensitized) across all time points of infection, 13 or more of the top 20 most pathways enriched in UPEC-infected versus mock-infected bladders were directly related to inflammation and immunity, including immune cell development and recruitment, cytokine expression and pattern recognition receptor signaling (*Supplementary file 1*). TNFα receptor 1 (TNFR1) and TNFα receptor 2 (TNFR2) signaling pathways were among the most highly upregulated pathways as determined by z-score during at least one time point for each mouse group (*Supplementary files 1 and 2*). TNFα is a cytokine involved in systemic inflammation, cell proliferation, and cell death. Notably, TNFα has been shown to be upregulated in C57BL/6 and CBA mice two hours after bladder infection (*Chan et al., 2013*; *Duell et al., 2012*), demonstrating that this response is not unique to C3H/HeN mice. Hierarchical clustering of the relative expression of genes known to be upregulated by TNFα signaling demonstrated the following patterns (*Figure 4*): all mock-infected mice (whether Adult Naive, Resolved, or Sensitized) clustered together (*Figure 4A*). At 3.5 hpi, 3 of 8 Resolved and 7 of 8 Sensitized mice infected with UPEC had evidence of TNFα pathway activation, clustering separately from mock-infected controls and from UPEC-infected Adult Naive mice. At 6 hpi, all Resolved (3 out of 3) and Sensitized (3 out of 3) mice, but only 1 out of 3 Adult Naive mice, infected with UPEC had evidence of strong TNFα pathway activation, clustering apart (far left side of dendrogram) from both mock-infected mice and the remaining UPEC-infected Adult Naive mice, of which the latter were intermediate in activation status (to the left of the mock-infected mice). At 24 hpi, strong TNFα signaling (far left clustering on the dendrogram) was present in 4 of 6 UPEC-infected Sensitized mice, but not in any Adult Naive or Resolved mice. Nonetheless, the remaining UPEC-infected Adult Naive and Sensitized mice had varying levels of activation at 24 hpi that clearly set them apart from both mock-infected controls and UPEC-infected Resolved mice, neither of which showed evidence of TNFR pathway activation at 24 hpi (*Supplementary file 2*). Pathway activation patterns were very similar between Resolved and Sensitized mice at 6 hpi. However, among the 100 most significantly enriched pathways at 24 hpi (*Supplementary file 1*), Resolved and Sensitized mice each possessed 37 pathways that were unique to their respective groups. Pathways unique to Sensitized mice were predominantly related to inflammation, whereas pathways unique to Resolved mice were related to a broader number of functions, such as cell growth and neuron ontogeny, but many of these signals were not strong enough to be assigned a z-score. Nevertheless, some pathways unique to Resolved mice are known to have anti-inflammatory effects, such as signaling pathways associated with derivatives of the omega-3-fatty acid docosahexaenoic acid (DHA), which can lead to production of factors such as resolvins that facilitate the resolution of inflammation (*Serhan, 2014*).

We also specifically analyzed the gene expression pattern of TNFα, along with cyclooxygenase-2 (COX-2, encoded by the *Ptgs2* gene), an enzyme critical in mediating urothelial inflammation during severe acute and recurrent cystitis (*Hannan et al., 2014*; *O'Brien et al., 2016*). The dynamics of *Ptgs2* expression closely mirrored that of *Tnf* expression and differed with infection history (*Figure 4B*). At 3.5 hpi, *Ptgs2* expression relative to mock-infected controls was significantly

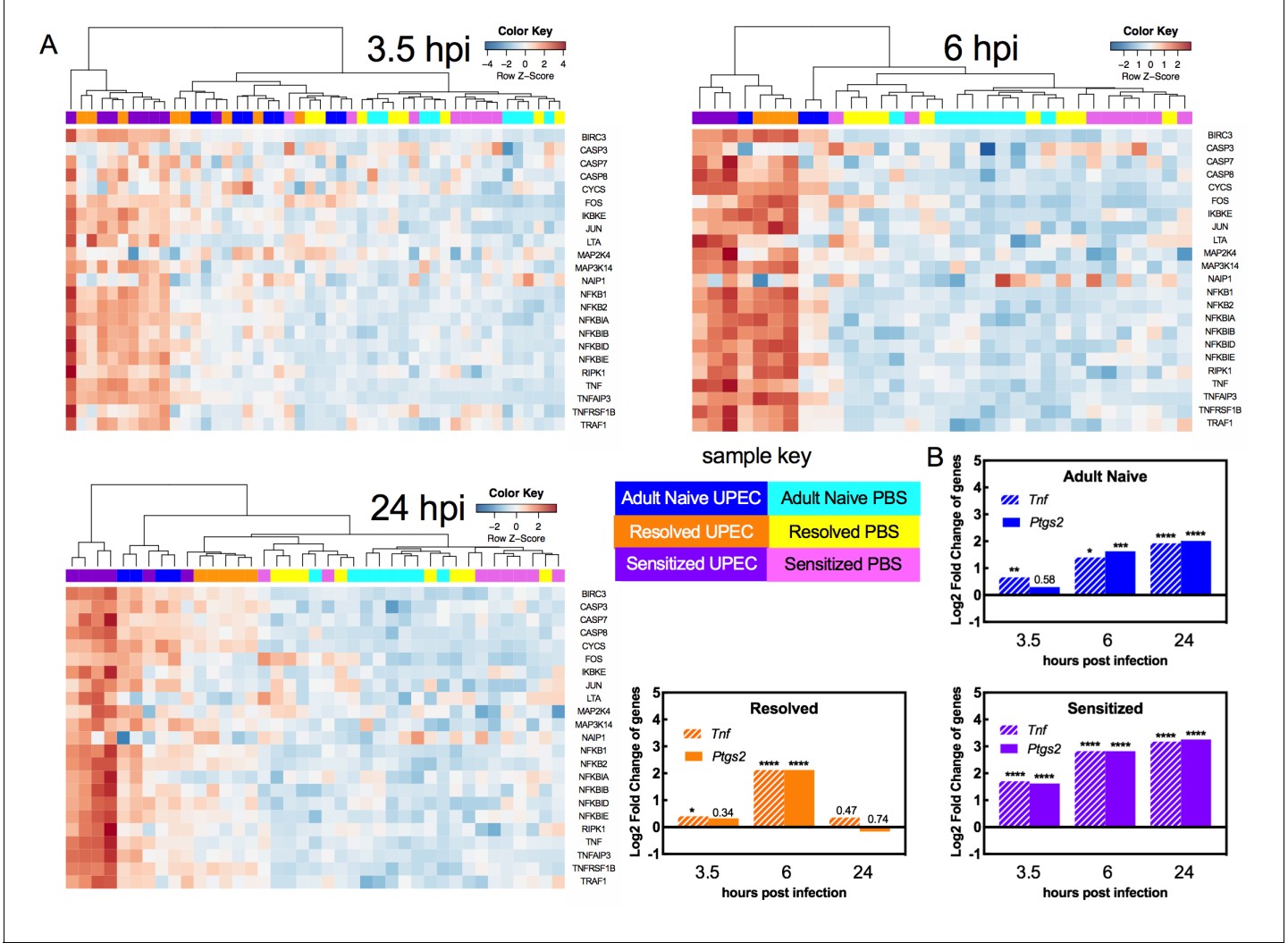

**Figure 4.** TNFα receptor pathway activation dynamics correlate with host susceptibility to recurrent cystitis. Pathway enrichment analysis and specific gene analysis was performed on the whole bladder RNA-seq data shown in *Figure 3*. (A) Heatmaps of expression of up-regulated genes in TNFα signaling pathways from Adult Naive, Resolved, and Sensitized mice at indicated time points. The dendrogram at the top of each heat map was produced by hierarchical clustering of gene expression. The same 3.5 hpi mock-infected controls are included as baseline controls for each heatmap. Specific fold changes and down-regulated genes are listed in *Supplementary file 2*. (B) Gene expression fold change of *Tnf* (encodes TNFα) and *Ptgs2* (encodes COX-2) in infected Adult Naive, Resolved, and Sensitized mice at indicated time points relative to mock-infected controls. Gene fold changes were estimated based on normalized counts using a shrinkage estimation model by DESeq2. Actual *P* values are indicated on the graphs if >0.05, or are represented by the following symbols: *p<0.05, **p<0.01, ***p<0.001, ****p<0.0001, Wald test, multiple comparison errors were corrected by Benjamini-Hochberg false-discovery rate correction. See also *Supplementary files 1* and *2*.

DOI: https://doi.org/10.7554/eLife.46677.009

The following source data is available for figure 4:

**Source data 1.** TNFα receptor pathway activation dynamics correlate with host susceptibility to recurrent cystitis.

DOI: https://doi.org/10.7554/eLife.46677.010

increased only in Sensitized mice. At 6 hpi, Adult Naive, Resolved, and Sensitized mice all showed evidence of bladder *Ptgs2* expression. At 24 hpi *Ptgs2* expression was further increased in Adult Naive mice and was sustained in Sensitized mice but fell to mock-infected levels in Resolved mice (*Figure 4B*). Thus, a robust early phase (first six hpi) TNFα signaling response is observed in C3H/HeN mice with a prior bladder infection, whether chronic (Sensitized) or self-limiting (Resolved). However, among previously infected mice (Sensitized and Resolved), TNFα signaling is sustained through 24 hpi only in Sensitized mice, who also experience high expression of bladder *Ptgs2*, which

is known to mediate mucosal immune damage to the urothelium and precipitate severe chronic and recurrent cystitis (*Hannan et al., 2010*; *O'Brien et al., 2016*).

## TNFα depletion increases bladder intracellular bacterial burdens in Resolved mice during acute rUTI

Previous studies in naive mice revealed that during infection, UPEC adhere to and invade into the superficial umbrella cells of the bladder epithelium, where they replicate and aggregate into intracellular bacterial communities (IBCs) within the cytosol, permitting the bacteria to increase in number and avoid innate immune defenses and the flow of urine (*Justice et al., 2004*; *Mulvey et al., 1998*; *Anderson et al., 2003*). We previously found that C3H/HeN mice that had experienced a prior UTI exhibit resistance to the stable formation of IBCs upon challenge infection: Resolved bladders harbored few if any IBCs at 6 hpi and none at 24 hpi, and no IBCs were observed in Sensitized bladders at either time point (*O'Brien et al., 2016*). To determine whether TNFα signaling during the early phase (first six hpi) of infection mediates this intracellular colonization resistance, we used a single dose of anti-TNFα antibody to deplete TNFα prior to challenge with $10^7$ cfu of UTI89. Bacterial colonization and IBC formation was compared to isotype-treated control mice in urine, bladders and kidneys, the latter because C3H/HeN mice are genetically prone to vesicoureteral reflux (*Murawski et al., 2010*). At 6 hpi, bladder bacterial burdens in Resolved mice given anti-TNFα antibody were increased relative to isotype-treated mice, while burdens in Adult Naive and Sensitized mice were unchanged (*Figure 5A*). Interestingly, TNFα depletion did not affect kidney colonization in any group (*Figure 5B*), but TNFα-depleted Resolved mice had lower urine titers at 6 hpi (*Figure 5C*). Due to the combination of higher bladder bacterial burdens and lower urine titers at 6 hpi, we hypothesized that TNFα depletion may have blocked the intracellular colonization resistance phenotype in Resolved mice, thus allowing IBC formation. Using fluorescence microscopy after infection with green fluorescent protein (GFP)-expressing UTI89, we found that TNFα depletion resulted in significantly more IBCs in Resolved bladders at 6 hpi, compared to isotype-treated controls (*Figure 5D*). In contrast, TNFα depletion did not affect IBC counts at 6 hpi in Adult Naive or Sensitized mice. In Resolved mice, bladder, but not kidney, bacterial burdens remained elevated in the TNFα-depleted group at 24 hpi (*Figure 5E and F*). By 7 days post-infection, urine titers were elevated in TNFα-depleted Resolved mice relative to isotype-treated mice (*Figure 5G*), suggesting that the increased IBC formation was allowing for more robust and longer-lasting infections. However, though TNFα depletion made Resolved mice more susceptible to acute cystitis upon challenge, a single dose of anti-TNFα antibody was insufficient to overcome their resistance to chronic infection, as 0 of 11 mice treated with anti-TNFα developed chronic cystitis lasting 4 weeks, compared to 0 of 11 mice treated with isotype. Nonetheless, TNFα depletion was sufficient to overcome intracellular colonization resistance during acute cystitis in Resolved, but not Sensitized mice.

## TNFα signaling restricts bladder intracellular colonization by UPEC in Resolved mice by promoting the exfoliation of infected bladder epithelial cells

We then sought to delineate how TNFα signaling restricts IBC formation in Resolved mice. We found that TNFα depletion did not affect the intracellular bladder bacterial burdens in Resolved mice at 3 or 4.5 hpi (*Figure 6A*), nor did it affect the number or morphology of IBCs present at 4.5 hpi (*Figure 6B* and *Figure 6—figure supplement 1*), indicating that TNFα signaling was neither limiting bacterial entry into urothelial cells nor affecting early IBC formation. We reasoned therefore that TNFα signaling was altering the inflammatory state of the bladder and thereby inducing exfoliation of the IBC-containing urothelial cells into the urine, as urothelial cell exfoliation is a known host defense mechanism against UPEC infection (*Mulvey et al., 1998*; *Choi et al., 2016*). Scanning electron microscopy (SEM) analysis of the urothelial surface of TNFα-depleted vs. control-treated Resolved mice at 6 hpi revealed striking differences: without TNFα depletion, clusters of neutrophils regularly gathered on the bladder surface, often underneath or around bulging umbrella cells that were likely ready to exfoliate, or in the gaps left by already exfoliated cells (*Figure 6C*), consistent with previous observations in UPEC-infected Resolved mice (*O'Brien et al., 2016*). With TNFα depletion, however, the bladder surface was minimally perturbed, with occasional bacteria on the surface and fewer neutrophils that were more scattered, rather than clustered, and little evidence of

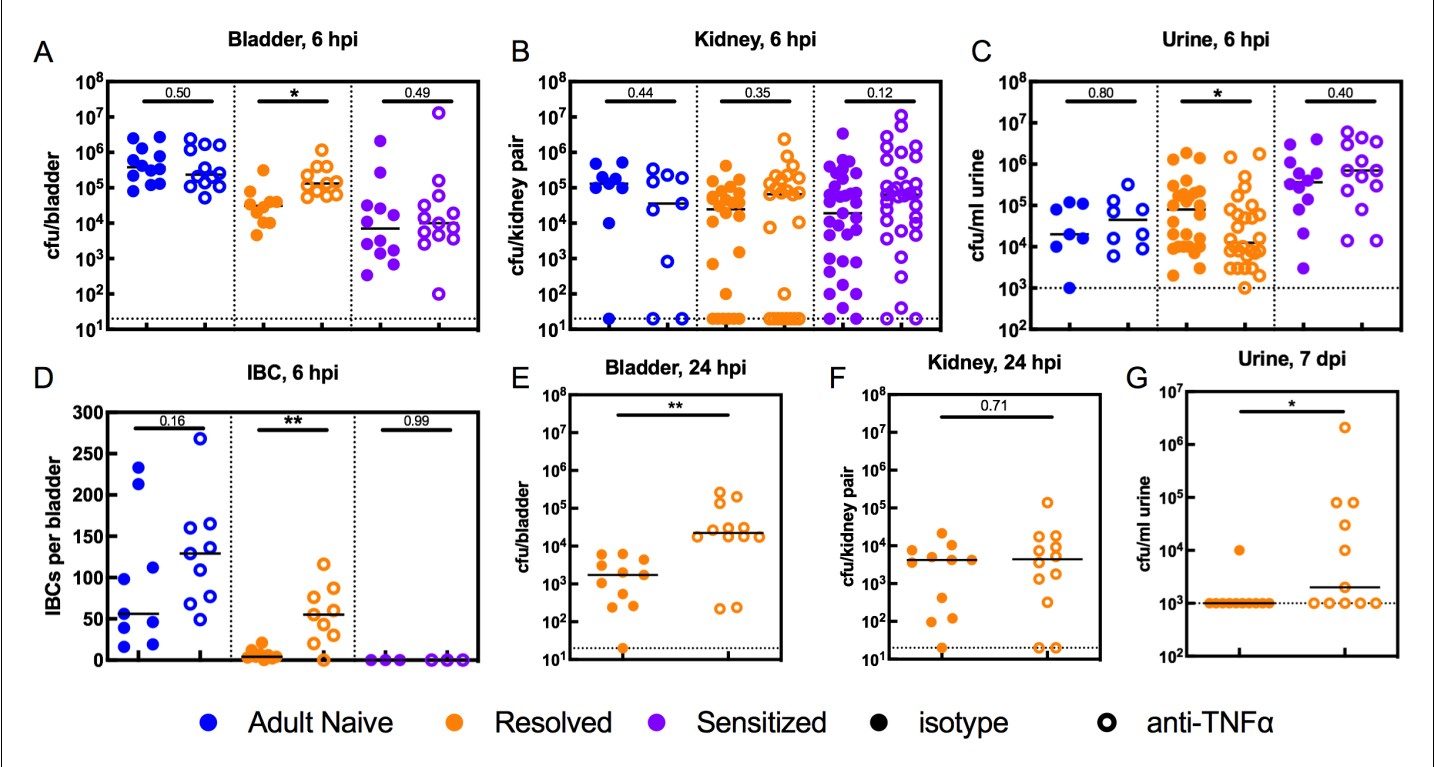

**Figure 5.** TNFα depletion increases intracellular bladder bacterial burdens in Resolved mice during acute rUTI. Adult Naive (blue symbols), Resolved (gold symbols), and Sensitized (purple symbols) mice were treated with anti-TNFα (open circles) or isotype control (solid circles) antibody 18 hr prior to infection with $10^7$ cfu UTI89 and sacrificed at indicated time points. Bacterial burdens were assessed at 6 hpi in the (**A**) bladder (cfu/bladder), (**B**) kidneys (cfu/kidney pair), and (**C**) urine (cfu/ml) in N = 5 independent experiments. (**D**) Intracellular bacterial communities (IBCs) were enumerated in N = 7 independent experiments. Bacterial burdens were enumerated at 24 hpi in Resolved mice for the (**E**) bladder (cfu/bladder) and (**F**) kidneys (cfu/kidney pair) in N = 2 independent experiments. (**G**) Urine bacterial burdens (cfu/ml) were measured in Resolved mice at 7 days post-inoculation (dpi); data from five independent experiments are depicted. Each data point represents the value for an individual mouse, bars indicate median values, negative results are plotted at the limit of detection (dotted line). Actual *P* values are indicated on the graphs if >0.05, or are represented by the following symbols: *p<0.05, **p<0.01, Mann-Whitney U test.

DOI: https://doi.org/10.7554/eLife.46677.011

The following source data is available for figure 5:

**Source data 1.** TNFα depletion increases intracellular bladder bacterial burdens in Resolved mice during acute rUTI.

DOI: https://doi.org/10.7554/eLife.46677.012

exfoliation (*Figure 6C*). Concordantly, bladder weights and neutrophils in the urine (pyuria) were each reduced in TNFα-depleted Resolved mice relative to isotype-treated mice at 6 hpi (*Figure 6D–E*).

Neutrophil recruitment to the bladder and subsequent transmigration into the lumen is associated with urothelial exfoliation and is required to prevent overwhelming infection (*Haraoka et al., 1999*), but excessive neutrophil-induced inflammation can increase the severity of acute cystitis and, correspondingly, the incidence of chronic cystitis by causing mucosal wounding (*Hannan et al., 2014*). To test whether TNFα depletion affected overall neutrophil recruitment to the bladder versus altering neutrophil transmigration across the urothelium, we performed flow cytometry of single cell suspensions generated from Resolved bladders at 6 hpi. TNFα-depleted mice had significantly fewer bladder-associated neutrophils than isotype-treated control mice (*Figure 6F*), indicating diminished neutrophil recruitment to the bladder in the absence of TNFα signaling. Thus, in C3H/HeN mice, a history of self-resolving UTI imparts the bladder with resistance to intracellular UPEC colonization that is mediated by bladder TNFα signaling, which promotes neutrophilic inflammation and urothelial exfoliation, thereby causing the shedding of IBC-containing cells.

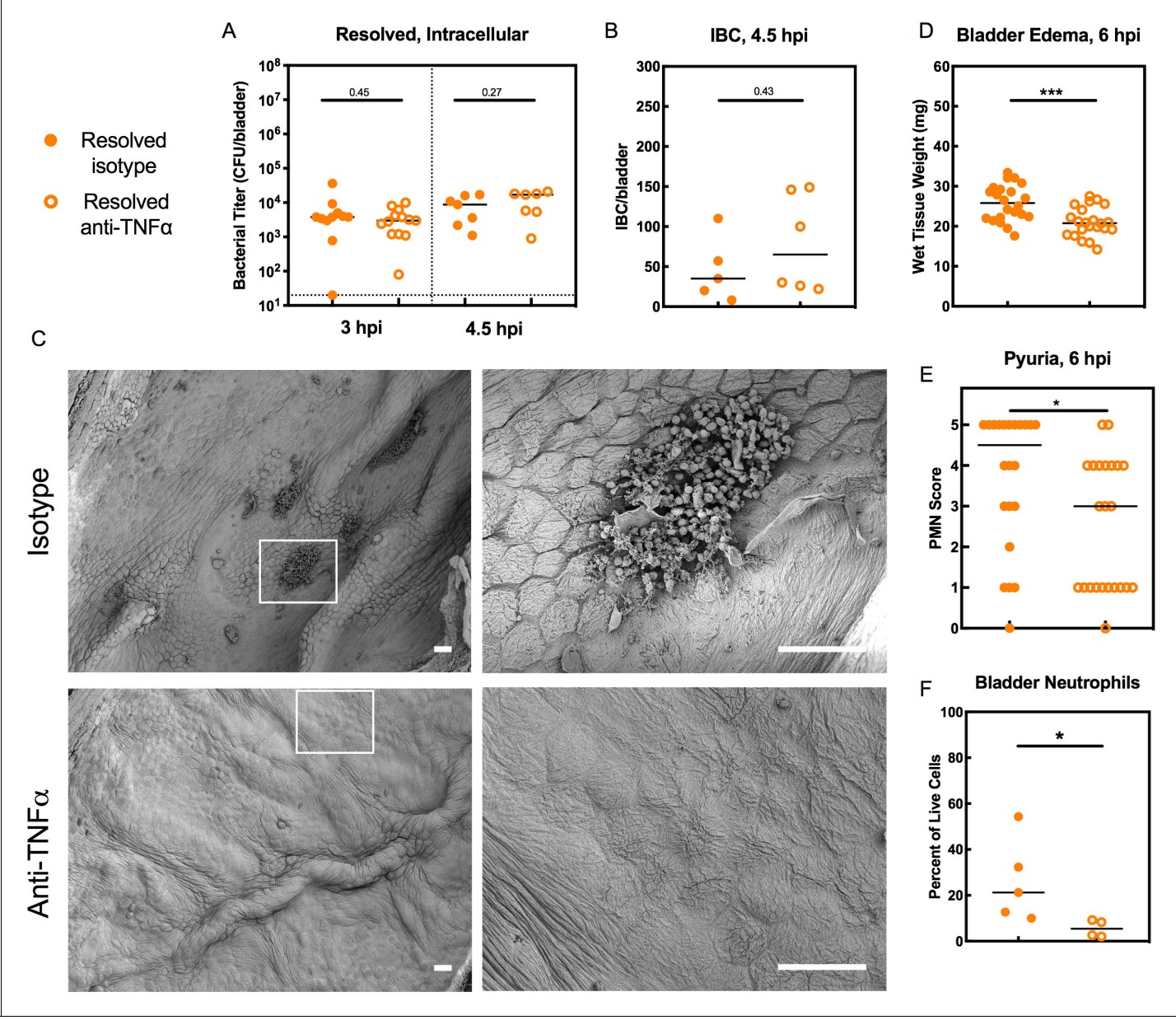

**Figure 6.** TNFα signaling restricts bladder intracellular colonization by UPEC in Resolved mice by promoting the exfoliation of infected bladder epithelial cells. Resolved mice were treated with anti-TNFα or isotype control antibody 18 hr prior to infection with $10^7$ cfu UTI89 and sacrificed at indicated time points. (**A**) Bladder intracellular bacterial burdens (cfu/bladder) of Resolved mice as determined by the ex vivo gentamicin protection assay in two independent experiments are shown at the indicated time points. (**B**) IBCs were enumerated in Resolved mice at 4.5 hpi with or without TNFα depletion from N = 2 experiments. See also *Figure 6—figure supplement 1*. (**C**) Scanning electron microscopy was used to assess the bladder luminal surface at 6 hpi in two independent experiments with n = 6 mice per group. Scale bars = 50 μm; white boxes outline areas visualized at a higher magnification to the right. At 6 hpi, (**D**) bladder edema and (**E**) pyuria were measured in N = 3 independent experiments. (**F**) Flow cytometry was performed on bladder single cell suspensions from Resolved mice in N = 2 independent experiments. Neutrophils were defined as CD11b[+], Ly6G[+], F4/80[-] cells. Data points represent value for each individual mouse, bars indicate median values, negative results are plotted at the limit of detection (dotted line). Actual *P* values are indicated on the graphs if >0.05, or are represented by the following symbols: *p<0.05, **p<0.01, Mann-Whitney U test.

DOI: https://doi.org/10.7554/eLife.46677.013

The following source data and figure supplement are available for figure 6:

**Source data 1.** TNFα signaling restricts bladder intracellular colonization by UPEC in Resolved mice by promoting the exfoliation of infected bladder epithelial cells.

DOI: https://doi.org/10.7554/eLife.46677.015

*Figure 6 continued on next page*

*Figure 6 continued*

**Figure supplement 1.** TNFα depletion did not affect IBC morphology in Resolved mice at 4.5 hpi.
DOI: https://doi.org/10.7554/eLife.46677.014

## TNFα depletion decreases the severity of acute rUTI in Sensitized mice

In Sensitized mice, TNFα depletion did not restore IBC formation at 6 hpi (*Figure 5D*), nor did it affect overall bladder bacterial burdens (*Figure 5A*), suggesting that colonization resistance during the first hours of challenge infection is unrelated to TNFα signaling in these mice. Previous work revealed that COX-2-mediated inflammation in the bladders of Sensitized mice during the first 24 hr of challenge infection overcame colonization resistance to allow severe acute and chronic rUTI, in part by promoting excessive neutrophil transmigration across the epithelium that caused severe bladder inflammation and mucosal wounding (*Hannan et al., 2014*; *O'Brien et al., 2016*). Treatment of Sensitized mice with a COX-2 inhibitor just prior to UPEC challenge protected against this outcome. Our findings that TNFα depletion in Resolved mice reduced neutrophil recruitment to the bladder as well as pyuria and urothelial exfoliation (*Figure 6*) suggest that sustained TNFα signaling in Sensitized mice may exacerbate acute bladder inflammation and mucosal wounding, predisposing to severe cystitis. We found that TNFα depletion in Sensitized mice resulted in a significant reduction in bladder bacterial burdens at 24 hpi (*Figure 7A*), with a decrease in median bladder titer from $3.8 \times 10^6$ cfu/bladder in isotype-treated mice to $8.4 \times 10^3$ cfu/bladder in TNFα-depleted mice. The effect of TNFα depletion in Sensitized mice was specific to the bladder, as kidney and urine bacterial burdens were unchanged (*Figure 7B–C*). TNFα depletion also significantly reduced bladder inflammation (edema) (*Figure 7D*), but, unlike with Resolved mice, did not affect the degree of pyuria observed *Figure 7E*. Thus, compared to Resolved mice, robust and sustained TNFα signaling in Sensitized mice induces prolonged inflammation through 24 hpi and promotes the development of severe acute cystitis.

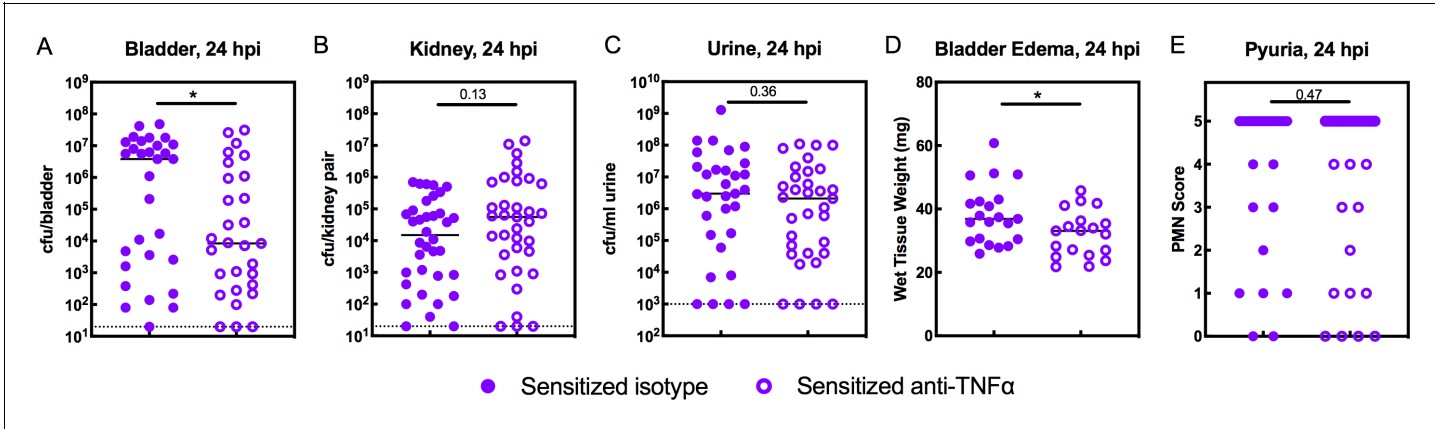

**Figure 7.** TNFα-depletion decreases the severity of acute rUTI in Sensitized mice. Sensitized mice were treated with anti-TNFα or isotype control antibody 18 hr prior to infection with $10^8$ cfu UTI89 infection and sacrificed 24 hpi. Bacterial burdens were enumerated in the (**A**) bladder (cfu/bladder), (**B**) kidneys (cfu/kidney pair), and (**C**) urine (cfu/ml urine), and (**D**) bladder weight and (**E**) pyuria were assessed in five independent experiments. Data points represent values for each individual mouse, bars indicate median values, negative results are plotted at the limit of detection (dotted line in A, B, and C). Actual *P* values are indicated on the graphs if >0.05, or are represented by the following symbols: *p<0.05. Mann-Whitney U test.
DOI: https://doi.org/10.7554/eLife.46677.016

The following source data is available for figure 7:

**Source data 1.** TNFα-depletion decreases the severity of acute rUTI in Sensitized mice.
DOI: https://doi.org/10.7554/eLife.46677.017

## Discussion

The mucosal response to pathogens is influenced by both the virulence of the infecting organism and the propensity for a given mucosa to respond to bacterial colonization, the latter being determined by both inherited and acquired host traits. In immunocompetent individuals, an initial infectious disease episode—particularly one that can lead to chronic inflammation, such as upper respiratory infections (sinusitis, tonsillitis, and nasopharyngitis) and urinary tract infections—can act as a potent effector of tissue remodeling that predisposes to recurrent infection (*Hooton et al., 1996*; *Nuhoglu et al., 2003*; *Toews, 2005*). In this work, we sought to understand the molecular basis for how the character of an initial mucosal infection may affect susceptibility to recurrent infection using an inbred immunocompetent mouse model of recurrent UTI. Based on our prior investigations of this model (*Hannan et al., 2010*; *Hannan et al., 2014*; *O'Brien et al., 2016*; *Schwartz et al., 2015*; *O'Brien et al., 2018*), we hypothesized that early host-pathogen interactions at the bladder mucosa trigger time-sensitive checkpoints that determine the outcome of infection, but that the character of a prior bladder infection can alter the kinetics and downstream effects of these disease checkpoints. By investigating the nature of acute inflammation during the first 24 hr of UPEC infection in isogenic mice that differed only in their disease histories, we identified distinct

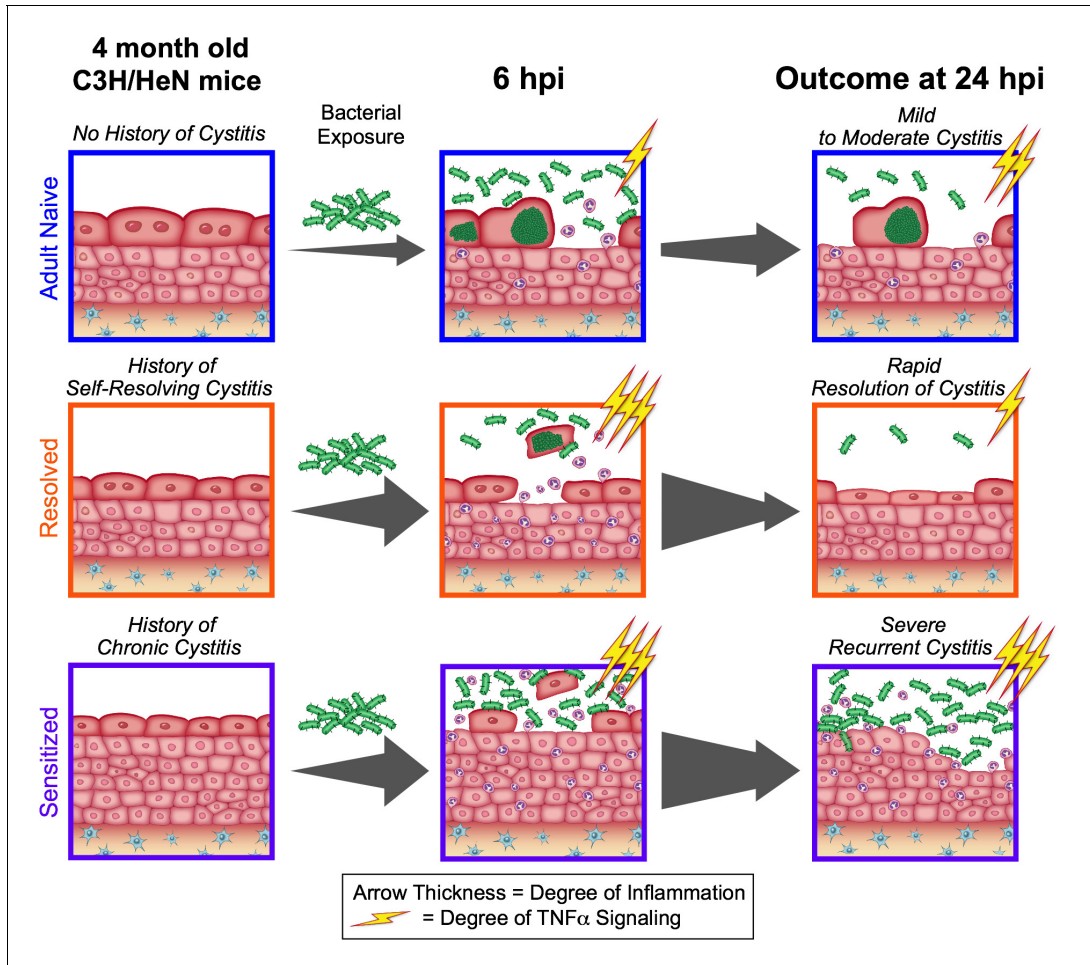

**Figure 8.** The outcome of a prior infection alters the balance towards either protection or susceptibility to recurrent infection through alterations in the dynamics of TNFα signaling. Upon bacterial challenge of Resolved mice, TNFα signaling (depicted as lightning bolts) is rapid, but transient (subsiding within the first 24 hr post-challenge), leading to rapid resolution of rUTI due to early exfoliation of infected bladder epithelial cells. However, when TNFα signaling is sustained (lasting 24 hr or more post-challenge), as occurs in Sensitized mice upon challenge, this contributes to mucosal wounding and severe rUTI.

DOI: https://doi.org/10.7554/eLife.46677.018

patterns of bladder inflammation that dictated the severity of recurrent cystitis (*Figure 8*). The nature of an initial bladder infection – self-resolving vs. long-lasting – imparts upon the bladder different capacities for TNFα signaling in response to recurrent infection, which in turn lead to different host outcomes: i) a history of a self-resolving infection corresponds with robust, but transient, early-onset TNFα signaling that leads to resistance to severe recurrent infection; whereas ii) a history of a long-lasting infection corresponds with robust early-onset TNFα signaling that is sustained though 24 hpi and leads to susceptibility to severe recurrent infection. Others have shown that TNFα plays a role in the bladder's response to UPEC infection in naive mice (*Engel et al., 2006*; *Schiwon et al., 2014*; *Gur et al., 2013*). However, our study demonstrates the centrality of this response and surprisingly shows that both the kinetics and the effects of this response can vary substantially based upon host infection history. These results have significance not only for our understanding of rUTI, but also of other recurrent mucosal infections.

Our RNA-seq and pathway analysis indicated that both TNFR1 and TNFR2 signaling pathways were activated in UPEC-infected bladders. TNFα binds to TNFR1 (expressed in most tissues, including urothelial cells) and/or TNFR2 (typically expressed by immune cells) and can activate distinct signaling cascades based on the receptors it binds to and the local microenvironment (*Wajant et al., 2003*; *Wang et al., 2018*; *Wang and Bjorling, 2011*). To add further complexity, both TNFα and the TNFRs can exist in either membrane bound or soluble forms. Both forms of TNFα can activate TNFR1, which leads either to NF-κB activation, cell survival, and cytokine production on the one hand, or cell death by apoptosis or necroptosis on the other (*Wang et al., 2018*). The balance between cell survival and cell death, as well as the inflammatory nature of the cell death in response to stimuli such as pathogen-associated molecular patterns (e.g. LPS), is influenced by many factors (*Geppert et al., 1994*). In contrast, only membrane-bound TNFα can activate TNFR2, thereby stimulating cell-mediated immunity, including activation of monocytes and lymphocytes, production of cytokines, and mediation of cytotoxic activity to promote host defense (*Horiuchi et al., 2010*). Although the pathway analysis employed in this study cannot distinguish between the two TNFR signaling pathways because of substantial overlap in the downstream regulated gene sets, it is likely that both TNFR1 and TNFR2 signaling are occurring in the bladder. Our use of whole bladder RNA-seq precludes us from identifying the cell types producing and responding to TNFα at this time.

The involvement of TNFα in both tissue damage and tissue regeneration represents an interesting functional duality (*Wajant et al., 2003*), and consequently, the host must tightly regulate TNFα activation to prevent damaging inflammation. We found that a major distinction between the responses of Sensitized and Resolved mice to bacterial challenge was the duration of TNFα signaling. Whereas Resolved mice had largely shut down TNFα signaling and bladder inflammation by 24 hpi, effectively nipping the infection in the bud, Sensitized mice had sustained TNFα signaling pathway activation in the bladder at 24 hpi, which contributed to the development of more severe infection. Adult Naive mice also had evidence of sustained bladder TNFα signaling pathway activation, though it was slower to develop and not as robust as in Sensitized mice at 24 hpi. The dynamics of bladder inflammation in Resolved mice—that is, robust early phase that is rapidly quenched by 24 hpi—has also been reported in juvenile naive C57BL/6 mice, which are naturally resistant to severe acute and chronic cystitis with UPEC (*Hannan et al., 2010*; *Chan et al., 2013*; *Duell et al., 2012*; *Mulvey et al., 1998*; *Ingersoll et al., 2008*; *Liu et al., 2015*). However, in contrast to our findings in Resolved mice, a prior study found no effect on overall neutrophil recruitment to the bladder in *juvenile naive* TNFR-/- vs WT C57BL/6J mice at 6 hpi, though transmigration of neutrophils across the bladder epithelium and exfoliation of infected bladder cells was impaired (*Schiwon et al., 2014*). While genotype can dictate these differences in the dynamics of host response in *naive* mice, our studies show that these dynamics are plastic and can be reshaped by non-germline factors such as prior infections and inflammation. Indeed, we previously demonstrated that superinfection of C57BL/6 mice can result in chronic cystitis, which subsequently sensitizes these mice to severe rUTI upon challenge (*O'Brien et al., 2016*).

We elucidated that previously infected C3H/HeN mice, regardless of disease outcome, shared an accelerated pattern of robust TNFα signaling and bladder inflammation relative to age-matched naive mice in the first 6 hours of infection (*Figure 8*). In naive mice, a critical event in bladder disease pathogenesis in the acute stages of *E. coli* infection is the formation of intracellular bacterial communities (IBCs), which allow the bacteria to rapidly increase in number while avoiding host defenses. However, previously infected C3H/HeN mice harbor very few, if any, IBCs at 6 hpi, a phenomenon

we have termed 'intracellular colonization resistance' (*O'Brien et al., 2016*). We found that in Resolved mice, abrogation of TNFα signaling disrupted intracellular colonization resistance by reducing the recruitment of neutrophils to the bladder and exfoliation of IBC-containing cells, thus resulting in abundant IBCs detected in the bladder epithelium at 6 hpi. TNFα induces downstream targets (e.g. neutrophil chemokines such as CXCL1) that stimulate neutrophil recruitment to the site of infection, supporting our observations of decreased neutrophil recruitment to the bladder and pyuria in TNFα-depleted Resolved mice. In contrast, Sensitized mice were unable to harbor IBCs in the presence or absence of TNFα signaling, likely due to the significantly smaller size of the superficial umbrella cells (average surface area approximately one eighth that of Adult Naive umbrella cells four weeks after the initiation of antibiotics), which likely restricts IBC formation (*O'Brien et al., 2016*; *O'Brien et al., 2018*). Thus, we found that the mechanism of intracellular colonization resistance differs between Sensitized and Resolved mice.

We found that the kinetics of TNFα expression and pathway activation mirrored the kinetics of *Ptgs2* (COX-2) expression regardless of disease history, suggesting that TNFα signaling may be associated with *Ptgs2* expression during bladder infection. COX-2 signaling is known to affect susceptibility to chronic and recurrent UTI in C3H/HeN mice, as COX-2-dependent severe inflammation and neutrophil transmigration through the bladder was shown to cause mucosal wounding leading to chronic cystitis (*Hannan et al., 2014*). Experimental suppression of COX-2-mediated acute inflammation with COX-2 inhibitors prevents chronic cystitis in naive mice and is a potent suppressor of recurrent UTI in Sensitized mice (*Hannan et al., 2014*; *O'Brien et al., 2016*). TNFα is capable of inducing COX-2 expression in some systems (*Chen et al., 2000*; *Mark et al., 2001*). However, whether COX-2 acts downstream of TNFα or both are activated by a common mechanism in mice during UPEC infection remains to be elucidated. While TNFα depletion in Sensitized mice phenocopied COX-2 inhibition in certain ways (e.g. reduced edema and reduced bacterial titers), unlike COX-2 inhibition, it did not affect pyuria, suggesting that neutrophil transmigration across the urothelium is independent of TNFα signaling in Sensitized mice. Taken together, these prior studies and our findings here elucidate an interesting dichotomy: whereas in Resolved mice, moderate, TNFα-mediated neutrophil recruitment to the bladder and subsequent exfoliation of IBC-containing urothelial cells allows for the clearance of infection, in Sensitized mice prolonged TNFα-signaling, coupled with COX-2 mediated transmigration of neutrophils across the bladder epithelium, exacerbates inflammation and mucosal wounding to promote severe acute cystitis. These different effects of TNFα signaling on the severity of recurrent cystitis may be a consequence of differences in epithelial remodeling that we previously demonstrated by quantitative proteomics (*Hannan et al., 2014*), and/or could be due to differences in bladder resident immune cell populations.

A question that this study was unable to answer is how the acute bladder inflammatory response is dampened so rapidly in Resolved mice. Although the RNA-seq pathway analysis gave some indication of activation of anti-inflammatory pathways in Resolved mice at 24 hpi, there was not a strong anti-inflammatory transcriptional signature that was also entirely absent in Sensitized mice. It may be that a unique, transient transcriptional signature arises in Resolved mice between the 6 and 24 hpi time points sampled, or alternatively, it may be that Resolved mouse bladders have additional inherent differences that are not evidenced by transcriptional changes. Previous ex vivo proteomics studies of urothelial cells isolated from convalescent mice suggested that the Sensitized urothelium is more susceptible to inflammatory cell death, neutrophil mediated damage, wounding, and oxidative stress (*Hannan et al., 2014*). Therefore, it is possible that the rapid elimination of intracellular bacterial communities in Resolved bladders, coupled with an inherent increased propensity for the Resolved urothelium to heal relative to the Sensitized urothelium, may explain the differences seen at 24 hpi. Understanding the specific mechanisms that allow Resolved mice to quickly quench bladder inflammation is an avenue of research that requires further investigation.

In summary, we show here that the nature and pattern of the bladder mucosal inflammatory response to UPEC infection, and particularly the dynamics of TNFα signaling, can be shaped by a prior infection in a way that dramatically alters host resistance to rUTI (*Figure 8*). Our studies indicate that TNFα signaling is a critical central mediator of bladder mucosal immune response to UPEC but can have different effects on acute disease outcome in previously infected mice depending upon the character of the prior infection. This central role is supported by clinical data indicating that prolonged use of anti-TNFα therapy in patients increases the risk of UTI (*Tong et al., 2015*). Understanding the underlying host mechanisms that dictate patient susceptibility to recurrent infections is

critical for developing effective new therapies that target the host inflammatory response as an alternative strategy to combat rapidly increasing antimicrobial resistance.

# Materials and methods

**Key resources table**

| Reagent type (species) or resource | Designation | Source or reference | Identifiers | Additional information |
|---|---|---|---|---|
| Strain, strain background (*Mus musculus*, female) | C3H/HeN | Envigo | C3H/HeN Hsd | |
| Strain, strain background (*Escherichia coli*) | UTI89 pANT4 | https://doi.org/10.1073/pnas.0308125100 (*Justice et al., 2004*) | | Episomal enhanced GFP; Kanamycin- and ampicillin-resistant |
| Strain, strain background (*Escherichia coli*) | UTI89 att$_{HK022}$::eGFP | https://doi.org/10.1128/IAI.73.11.7657-7668.2005 (*Wright et al., 2005*) | UTI89-Kan$^R$ | Chromosomal enhanced GFP; Kanamycin-resistant |
| Strain, strain background (*Escherichia coli*) | UTI89 attλ::PSSH10-1 | https://doi.org/10.1128/IAI.73.11.7657-7668.2005 (*Wright et al., 2005*) | UTI89-Spc$^R$ | Spectinomycin-resistant |
| Antibody | Anti-TNFα (rat IgG1 monoclonal, clone XT3.11) | BioXCell | BP0058, RRID:AB_1107764 | 10 mg/kg IP |
| Antibody | Anti-HRP isotype control (rat IgG1 monoclonal, clone XT3.11) | BioXCell | BP0088, RRID:AB_1107775 | 10 mg/kg IP |
| Antibody | Anti-mouse CD11b-PE conjugate (rat monoclonal, clone M1/70) | BD Biosciences | 553311, RRID:AB_394775 | (1:200) |
| Antibody | Anti-mouse Ly6G-FITC conjugate (rat monoclonal, clone 1A8) | BioLegend | 127605, RRID:AB_1236488 | (1:200) |
| Antibody | Anti-mouse F4/80-APC conjugate (rat IgG1 monoclonal, clone BM8) | BioLegend | 123115, RRID:AB_893493 | (1:200) |
| Antibody | Anti-mouse CD16/32, FcR block (rat monoclonal) | Biolegend | 101301, RRID:AB_312800 | |
| Commercial assay or kit | Duoset IL-6 ELISA kit | R and D Systems | DY406 | |
| Commercial assay or kit | Duoset CXCL1 ELISA kit | R and D Systems | DY453 | |
| Commercial assay or kit | Duoset CCL2 ELISA kit | R and D Systems | DY479 | |
| Commercial assay or kit | RNeasy Plus kit | Qiagen | 74136 | |
| Commercial assay or kit | RiboZero rRNA depletion kit | Illumina | MRZG12324 | |

*Continued on next page*

*Continued*

| Reagent type (species) or resource | Designation | Source or reference | Identifiers | Additional information |
|---|---|---|---|---|
| Commercial assay or kit | SMARTScribe reverse transcriptase | Clontech | 639536 | |
| Software, algorithm | Salmon | Salmon | 0.8.2, RRID:SCR_017036 | |
| Software, algorithm | DESeq2 | DESeq2 | 1.14.0, RRID:SCR_015687 | |
| Software, algorithm | Ingenuity Pathway Analysis | Qiagen Bioinformatics | RRID:SCR_008653 | |
| Other | Wheat germ agglutinin, AlexaFluor 594 conjugate | ThermoFisher | W11262 | (1:1,000) |
| Other | DAPI stain | ThermoFisher | D1306 | (1:20,000) |
| Other | Prolong Gold Anti-fade | ThermoFisher | P36930 | |

## Ethics statement

All animal experimentation was conducted according to the National Institute of Health guidelines for the housing and care of laboratory animals and performed in accordance with institutional regulations after review and approval by the Institutional Animal Care and Use Committee (animal protocol number 20180276) at Washington University in St. Louis, Missouri (Office of Laboratory Animal Welfare (OLAW) Assurance number A3381-01).

## Bacterial strains

The UPEC strain primarily used in this study was a kanamycin-resistant derivative of the human cystitis isolate UTI89 (*Mulvey et al., 2001*): UTI89 *att_{HK022}::COM-GFP* (UTI89-*Kan^R*) (*Wright et al., 2005*). For UPEC challenge of previously infected mice, we used a spectinomycin-resistant derivative: UTI89 *attλ::PSSH10-1* (UTI89-*Spc^R*) (*Wright et al., 2005*). For enumeration of IBCs by epifluorescence microscopy, we used UTI89 *pANT4*, which contains a plasmid that constitutively expresses eGFP. Bacteria were routinely cultured in lysogeny broth (LB).

## Mouse infections

Female C3H/HeN mice were purchased from Envigo (Indianapolis, IN). Bacterial strains were inoculated into 20 ml of LB directly from freezer stock, grown statically at 37°C overnight, and sub-cultured 1:1000 into 20 ml of fresh LB and again grown statically at 37°C for 18 hr. These cultures were spun at room temperature for 10 min at 3000 xg, re-suspended in 10 ml phosphate-buffered saline, pH = 7.4 (PBS), and diluted to approximately $2–4 \times 10^9$ cfu/ml (OD$_{600}$ = 3.5). 50 μl of this suspension (~$1–2 \times 10^8$ cfu) was inoculated into the bladders of 7–8 week old female Juvenile Naive mice by transurethral catheterization. All initial infections and most challenge infections were performed with a high-dose inoculum of $10^8$ cfu UTI89 to reduce variability in infection dynamics and host responses (*Hannan et al., 2010*). For TNFα depletion studies in *Figures 5* and *6, a* $10^7$ cfu UTI89 inoculum was used to keep consistent with COX-2 inhibitor treatment studies described previously (*O'Brien et al., 2016*).

## IBC enumeration

Bladders were aseptically removed and hemisected. Each hemisphere was splayed out on silica plates using forceps and pins. Bladders were fixed with 4% paraformaldehyde (room temperature, 1 hr, shaking), washed with PBS, and incubated with 0.01% Triton-X in PBS (10 min). Bladders were stained with wheat germ agglutinin Alexa Fluor 594 conjugate (Molecular Probes) and 4',6-diamidino-2-phenylindole DAPI (Life Technologies), washed with PBS, and mounted on slides with Prolong Gold Anti-Fade (ThermoFisher). Images of bladder hemispheres were captured on a Zeiss Axio Imager M2 upright wide-field fluorescence microscope or a Zeiss Observer D1 inverted wide-field

fluorescence microscope and the number of IBCs across the entire interior of the bladder was enumerated manually by ImageJ (NIH, ImageJ bundled with Java 1.8.0_101).

## Tissue bacterial enumeration

At the indicated time points, mice were humanely euthanized and bladders were aseptically harvested and homogenized in PBS using a FastPrep-24 bead beater (MP Biomedicals, Santa Ana, CA). Homogenates were then serially diluted in PBS and spotted onto LB agar plates with and without antibiotic selection. Bladder intracellular bacterial burdens were determined using an ex vivo gentamicin protection assay performed as previously described (*Mulvey et al., 1998*) with the following modifications: after washing three times with PBS, bladders were incubated at 37°C for 75 min in gentamicin in RPMI cell culture medium with no serum added.

## Urine collection, bacterial enumeration, and urine sediment analysis

Urines were collected by applying suprapubic pressure with proper restraint and collecting the urine stream in sterile 1.5 ml Eppendorf tubes. Urines were then serially diluted in PBS and 10 µl total of each dilution was spotted onto LB and LB with 25 µg/ml kanamycin (LB/Kan25) agar plates. Urine sediments were obtained by cyto-centrifuging 80 µl of a 1:10 dilution of the collected urine onto poly-L-lysine-coated glass slides and stained as described (*Rosen et al., 2007*). To assess pyuria, stained urine sediments were examined by light microscopy on an Olympus BX51 light microscope (Olympus America), and the average number of polymorphonuclear leukocytes (PMN) per 400x magnification field (hpf) was calculated from counting five fields. A semi-quantitative scoring system of 0–5 was modified from an earlier study to facilitate pyuria analysis: 0, less than 1 PMN/hpf; 1, 1–5 PMN/hpf; 2, 6–10 PMN/hpf; 3, 11–20 PMN/hpf; 4, 21–40 PMN/hpf, and 5,>40 PMN/hpf (*Hannan et al., 2010*; *Hannan et al., 2014*).

## Recurrent UTI model

Mice were initially inoculated with either $10^8$ cfu of UTI89-$Kan^R$ (convalescent) or PBS (Adult Naive) and longitudinal urinalysis was performed to determine disease outcome. Urines were collected at 1, 3, 7, 10, 14, 21, and 28 days post-infection (dpi) and mice with persistent high-titer bacteriuria, which we define as the presence of $>10^4$ cfu/mL of UTI89-$Kan^R$ at every time point, were deemed to have chronic cystitis (Sensitized), and all other UPEC-infected mice were deemed to have resolved the infection (Resolved). At 28 dpi, all mice were treated with trimethoprim and sulfamethoxazole in the drinking water daily for 10 days at concentrations of 54 and 270 µg/ml, respectively (*Schilling et al., 2002*). During this time, longitudinal urinalysis was continued weekly to confirm clearance of bacteriuria. Four to six weeks after the initiation of antibiotic therapy mice were challenged with $10^7$ or $10^8$ cfu of UTI89-$SpcR$. Mice were humanely euthanized at various time points and tissue titers determined as above.

## TNFα depletion in mice

Mice were administered anti-TNFα antibody (clone XT3.11) or isotype control (clone HRPN) (Bio-XCell, West Lebanon, NH) via intraperitoneal injection 18 hr before infection. Given the recommended dosage of 10 mg/kg, mice were administered 300 µg. Treatment was randomly assigned to each cage of mice.

## Inflammation profiling

Mice were infected with $10^8$ cfu of UPEC or PBS and were humanely euthanized at indicated time points. Bladder edema was determined obtained by weighing bladders in pre-weighed Eppendorf tubes and calculating bladder weight. Bladder homogenates were obtained via bead beating as described above, and after cfu plating, the remaining homogenates were spun down at 15,000 rpm for 5 min at 4°C and the supernatants were stored at −80°C. Cytokine expression in bladder homogenate supernatants was measured by enzyme-linked immunosorbent assay (ELISA) using DuoSet kits (R and D Systems, DY406, DY453, DY479) following the manufacturer's instructions, with the following modifications: incubation with the capture antibody was lengthened to overnight at 4°C, and washes between incubations were performed five times instead of 3.

## Histopathology and immunofluorescence

Mice were infected with $10^8$ cfu of UPEC or PBS and were humanely euthanized at indicated time points. For histopathology assessment, bladders were excised, fixed with Methacarn (60% methanol, 30% chloroform, 10% glacial acetic acid), embedded in paraffin blocks, and cut into 5 µm sections. Hematoxylin and eosin (H and E) staining was performed and bladder sections were scored in a blinded fashion by a veterinarian using a semi-quantitative scoring system: 0 = normal, 1 = subepithelial cell inflammatory infiltration (focal and multifocal), 2 = edema and subepithelial inflammatory cell infiltration (diffuse), 3 = marked subepithelial inflammatory cells with necrosis and neutrophils in and on bladder mucosal epithelium, 4 = inflammatory cell infiltrate extends into muscle in addition to criteria for grade 3, 5 = loss of surface epithelium (necrosis with full-thickness inflammatory cell infiltration) (*Hopkins et al., 1998*).

## Flow cytometry

Mice were infected with $10^7$ cfu UPEC and were humanely euthanized at the indicated time point. Bladders were harvested and made into single-cell suspension via collagenase IV/DNase I digestion for 90 min at 37°C and passing through a 40 µm filter as previously described (*Ingersoll et al., 2008*). Cell surface markers were stained with fluorochrome-conjugated monoclonal antibodies (MAbs) (F4/80, CD11b, Ly6G from Biolegend and BD Biosciences) in FcR block (Biolegend). Cells were counterstained with propidium iodide (PI) prior to analysis on a FacsCalibur flow cytometer (BD Biosciences). Neutrophils were identified according to the following surface marker profile: $CD45^+$ $CD11b^+$ $Ly6G^+$ $F4/80^-$ and relative abundance was reported as a percentage of live ($PI^-$) cells. Data were analyzed with FlowJo software version 10.

## Scanning electron microscopy

Mice were infected with $10^8$ cfu UPEC or PBS. Bladders were fixed in their native state (prior to harvest) using the 'balloon method' (*Walker et al., 2017*). Briefly, the body cavity was opened, a catheter was inserted transurethrally and bladders were inoculated with 50 ul freshly prepared SEM fixative (2% paraformaldehyde, 2% glutaraldehyde in 0.1 M cacodylate, pH 7.4, warmed to 37°C). The bladder was clamped with a hemostat before the catheter was withdrawn, and then excised and placed in 1 ml EM fixative overnight. Bladders were then bisected, post-fixed in 1.0% osmium tetroxide, dehydrated in increasing concentrations of ethanol, further dehydrated at 31.1°C and 1072 p.s.i. for 16 min in a critical point dryer, then sputter-coated with 10 nm iridium and imaged on a Zeiss Crossbeam 540 FIB-SEM or a Zeiss Merlin FE-SEM.

## RNA isolation

Mice infected with $10^8$ cfu of UPEC or PBS were humanely euthanized at indicated time points. All mock-infected mice were euthanized at 24 hpi. Bladders were aseptically harvested, flash-frozen in liquid nitrogen, and stored at −80°C. RNA isolation from whole bladders was performed using the with QIAGEN RNeasy Plus kit (74136). RNA quality was spot-checked using the Bioanalyzer platform (Agilent) and all RNA integrity number (RIN) scores were 8.9 or higher.

## Generation and analysis of RNA-Seq data

Illumina cDNA libraries were generated using a modified version of the RNAtag-seq protocol (*Shishkin et al., 2015*). Briefly, fragmented, dephosphorylated total RNA was ligated to barcoded DNA adapters carrying 5'-AN$_8$-3' barcodes and a 3' blocking group. Barcoded RNAs were pooled, depleted of rRNA using RiboZero (Illumina MRZG12324), then reverse transcribed using SMART-Scribe (Clontech) as described to add an adapter to the 3' end of the cDNA by template switching (*Zhu et al., 2001*). Addition of Illumina P5 and P7 sequences to these cDNAs was achieved by PCR using tailed oligos targeting adapter sequences (*Supplementary file 3*). The resulting libraries were sequenced on the Illumina Nextseq 2500 platform to generate paired end reads. Samples were processed for library construction and sequencing in two batches: PBS mock-infected and UPEC-infected bladder RNA samples taken at 3.5 hpi were included in batch one; whereas UPEC-infected 6 and 24 hpi bladder RNA samples were processed 5 months later using the same reagents in batch two. Sequencing reads from each sample in a pool were demultiplexed based on their associated barcode sequence using custom scripts (https://github.com/broadinstitute/split_merge_pl; copy

archived at https://github.com/elifesciences-publications/split_merge_pl) (*Bandyopadhyay, 2019*). Up to one mismatch in the barcode was allowed provided it did not make assignment of the read to a different barcode possible. Barcode sequences were removed from the first read as were terminal G's from the second read that may have been added by SMARTScribe during template switching. Reads were aligned to *Mus musculus* Ensembl sequence GRCm38r94p6 mm10 using bbmap_37.10 (https://jgi.doe.gov/data-and-tools/bbtools/) and read counts were assigned to annotated transcripts using Salmon_0.8.2 (*Patro et al., 2017*). Differential expression and pathway analysis was conducted with DESeq2_1.14.0 (*Love et al., 2014*) and Ingenuity Pathway Analysis (https://www.qia-genbioinformatics.com/), respectively. Sequencing data were deposited in GEO database, accession number: GSE117532 (https://www.ncbi.nlm.nih.gov/geo/query/acc.cgi?acc=GSE117532).

## Statistical analysis

1) RNA-seq: DESeq2 assumes a negative binomial distribution for gene counts, normalizes for read depths and fits a generalized linear model. Statistical significance in gene expression differences were assessed by the Wald test and multiple comparison errors were corrected by Benjamini-Hochberg false-discovery rate correction ($P_{adjusted}$). $P_{adjusted}$ <0.05 was used as the cutoff for significantly differentially expressed genes without a fold-change cutoff. IPA employs a right-tailed Fisher's exact test, with $P_{adjusted}$ <0.05 deemed significantly enriched. A fold change cutoff of 2 was used in IPA analysis. Principal component analyses were performed with DESeq2 rlog counts using default settings in DESeq2 and including batch number as a variable to correct for any batch effect. 2) Microbiology, immunology, and pathology: statistical significance of bladder, urine, and kidney bacterial titers, histopathology scores, bladder edema, cytokine expression level, pyuria, IBC counts, and flow cytometry were assessed by Mann-Whitney U when comparing two groups or Kruskal-Wallis test with Dunn's correction for multiple comparisons when comparing three or more groups. All the above items used two-tailed tests. 3) For all experiments, each data point came from an individual mouse, which is our definition of a biological replicate. For ELISA experiments, two technical replicates (bladder supernatant from the same mouse were applied to two wells of the ELISA plate) were assayed per biological replicate and the average value was plotted. 4) No specific computation of sample size was applied beforehand. Sample size for each experiment was determined a priori and following convention in the field: 3–7 mice per group per replicate experiment, with a minimum of 2–3 replicate experiments (except for RNAseq experiments).

## Acknowledgements

The authors thank Danielle Liu for technical assistance and Karen Dodson for editorial assistance. This work was supported by the National Institutes of Health (R01 DK51406 to SJH, U01 AI95542 to MC, SJH and TJH, K08 AI083746 to TJH); the Office of Research on Women's Health Specialized Center of Research (P50 DK64540), URL: https://orwh.od.nih.gov; a National Institutes of Health Mucosal Immunology Studies Team consortium Young Investigator Award (U01 AI095776 to TJH), URL: https://www.mucosal.org; and the National Science Foundation (Graduate Research Fellowship #DGE–114395 to VPO), URL: https://www.nsfgrfp.org/. As well, scanning electron microscopy sample preparation and imaging was conducted at the Washington University Center for Cellular Imaging (WUCCI), which is supported by the Washington University School of Medicine, The Children's Discovery Institute of Washington University and St. Louis Children's Hospital (CDI-CORE-2015–505), and the Foundation for Barnes-Jewish Hospital (3770). RNA-seq analysis design and support was provided by the Rheumatic Diseases Research Resource-based Center at Washington University (EDOR was partially supported by NIH/NIAMS grant P30-AR073752) and The Broad Institute of Massachusetts Institute of Technology and Harvard (JL has been funded in part with Federal funds from the National Institute of Allergy and Infectious Diseases, National Institutes of Health, Department of Health and Human Services, under Grant Number U19AI110818 to the Broad Institute). The funders had no role in study design, data collection and analysis, decision to publish, or preparation of the manuscript.

# Additional information

## Funding

| Funder | Grant reference number | Author |
|---|---|---|
| National Institute of Diabetes and Digestive and Kidney Diseases | R01 DK51406 | Scott Hultgren |
| National Institute of Allergy and Infectious Diseases | U01 AI95542 | Marco Colonna Scott Hultgren Thomas J. Hannan |
| National Institute of Allergy and Infectious Diseases | K08 AI083746 | Thomas J. Hannan |
| NIH Office of the Director | P50 DK64540 | Scott Hultgren |
| National Institute of Allergy and Infectious Diseases | U01 AI095776 | Thomas J. Hannan |
| National Science Foundation | Graduate Research Fellowship #DGE-114395 | Valerie Phoebe O'Brien |
| National Institute of Arthritis and Musculoskeletal and Skin Diseases | P30 AR073752 | Elisha D.O. Roberson |
| National Institute of Allergy and Infectious Diseases | U19 AI110818 | Jonathan Livny |

The funders had no role in study design, data collection and analysis, decision to publish, or preparation of the manuscript.

## Author contributions

Lu Yu, Valerie P O'Brien, Conceptualization, Formal analysis, Investigation, Methodology, Writing—original draft, Writing—review and editing; Jonathan Livny, Conceptualization, Resources, Formal analysis, Investigation, Methodology, Writing—review and editing; Denise Dorsey, Investigation; Nirmalya Bandyopadhyay, Software, Formal analysis, Investigation; Marco Colonna, Conceptualization, Resources, Funding acquisition; Michael G Caparon, Supervision, Writing—review and editing; Elisha DO Roberson, Resources, Formal analysis, Methodology, Writing—review and editing; Scott J Hultgren, Conceptualization, Resources, Supervision, Funding acquisition, Writing—review and editing; Thomas J Hannan, Conceptualization, Formal analysis, Supervision, Funding acquisition, Investigation, Methodology, Writing—original draft, Writing—review and editing

## Author ORCIDs

Lu Yu (iD) https://orcid.org/0000-0002-2361-2347
Valerie P O'Brien (iD) https://orcid.org/0000-0003-0502-2690
Elisha DO Roberson (iD) https://orcid.org/0000-0001-5921-2399
Scott J Hultgren (iD) https://orcid.org/0000-0001-8785-564X
Thomas J Hannan (iD) https://orcid.org/0000-0003-0498-8231

## Ethics

Animal experimentation: All animal experimentation was conducted according to the National Institute of Health guidelines for the housing and care of laboratory animals and performed in accordance with institutional regulations after review and approval by the Institutional Animal Care and Use Committee (animal protocol number 20180276) at Washington University in St. Louis, Missouri (Office of Laboratory Animal Welfare (OLAW) Assurance number A3381-01).

## Decision letter and Author response

Decision letter https://doi.org/10.7554/eLife.46677.026

Author response https://doi.org/10.7554/eLife.46677.027

## Additional files

### Supplementary files

• Supplementary file 1. The enriched pathways in Adult Naive, Resolved, and Sensitized mice infected with UPEC for 3.5, 6, or 24 hpi, compared to mock infection. Gene fold change cut off for use in IPA analysis is 2-fold or greater; cut off for $P$ value of enrichment is 0.05 or -$\log_{10}$($P$ value) of 1.3; 'ratio' is the number of genes differentially expressed in the samples divided by the number of genes in this pathway; and Z-score is a statistical measure of the match between expected relationship direction and observed gene expression: positive z-score predicts activation, negative z-score predicts suppression.
DOI: https://doi.org/10.7554/eLife.46677.019

• Supplementary file 2. Genes regulated by TNFα signaling in Adult Naive, Resolved, and Sensitized mice infected with UPEC for 3.5, 6, or 24 hpi, compared to mock infection. 'Expr Log Ratio"indicates expression $\log_2$ fold change; 'Expr p-value' indicates expression-adjusted $P$ value; and the cut off for $P$ value of enrichment is 0.05 or -$\log_{10}$($P$ value) of 1.3.
DOI: https://doi.org/10.7554/eLife.46677.020

• Supplementary file 3. Details of library construction and sequencing.
DOI: https://doi.org/10.7554/eLife.46677.021

• Transparent reporting form
DOI: https://doi.org/10.7554/eLife.46677.022

### Data availability

Sequencing data were deposited in GEO database under accession number GSE117532.

The following dataset was generated:

| Author(s) | Year | Dataset title | Dataset URL | Database and Identifier |
|---|---|---|---|---|
| Yu L, O'Brien VP, Livny J, Dorsey D, Bandyopadhyay N, Colonna M, Caparon MG, Roberson EDO, Hultgren SJ, Hannan TJ | 2018 | Host susceptibility to recurrent cystitis is shaped by bladder TNF-alpha signaling dynamics | https://www.ncbi.nlm.nih.gov/geo/query/acc.cgi?acc=GSE117532 | NCBI Gene Expression Omnibus, GSE117532 |

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
