## [Decision Letter]

Thank you for submitting your article "Mucosal infection rewires TNFα signaling dynamics to skew susceptibility to recurrence" for consideration by *eLife*. Your article has been reviewed by three peer reviewers, and the evaluation has been overseen by a Reviewing Editor and Wendy Garrett as the Senior Editor. The following individual involved in review of your submission has agreed to reveal their identity: Mark A Schembri.

The reviewers have discussed the reviews with one another and the Reviewing Editor has drafted this decision to help you prepare a revised submission.

In the present study, you elaborate on a strong model to study recurrent UTI to study how the outcome of a first infection impacts the outcome of a new one. Briefly, C3H/HeN mice initially exposed to UPEC can spontaneously resolve or develop a chronic cystitis requiring antibiotic treatment, each outcome predisposing a new infection to be mild or acute, respectively. You carried out a 24hr time course transcriptomics analysis on Naïve/Resolved/Sensitized UPEC infected and Naïve/Resolved/Sensitized mock mice allowing you to present strong evidence to show that TNFα plays a central role in this response. The robust early immune response (6hpi) was characterized by an increase in TNFα and Cox-2 signaling that resulted in increased neutrophil recruitment and urothelium exfoliation. However, those mice that had previously resolved a UTI were able to dampen this immune response by 24 hours, while sensitized mice were not, leading to enhanced inflammation and increased severity of infection.

The reviewers all agreed that the study is very well conducted and that findings are exciting with the potential to improve understanding of other mucosal infections that are also associated with high rates of recurrence. They deem the results convincingly supported by the experimental data, but request to see a few revisions to your work before it can be accepted for publication.

Essential revisions:

1) How do resolving mice dampen or control the TNFα response is an unanswered question. What differs between the two groups that allows resolving mice to dampen the response, but leaves sensitized mice activated? Did the RNA-seq allow to identify any pathways at 24 hour that might provide a clue?

2) Figure 3: Some technical aspects on the transcriptomics are missing like the RNA quality (to be added in the Materials and methods) and a supplementary file giving the detail of all the libraries generated (with the barcode associated).

3) Figure 3: It would be interesting to compare the results of 'adult Naïve PBS', 'Sensitized PBS' and 'Resolved PBS' with a PCA analysis. Since these are the baseline experiments, one would like to understand how the initial conditions vary between the different group of mice.

4) Figure 3: Similar experiments have been reported by the authors previously – in the supplementary materials, is it possible to compare the present results with those published before?

5) Figure 3: please integrate a PCA that compiles all the experiments to have a global view of the data.

6) Please include a model in the discussion detailing the disease progression for each group (Resolving vs. Sensitized).

---

## [Author Response]

Essential revisions:1) How do resolving mice dampen or control the TNFα response is an unanswered question. What differs between the two groups that allows resolving mice to dampen the response, but leaves sensitized mice activated? Did the RNA-seq allow to identify any pathways at 24 hour that might provide a clue?

Unfortunately, despite our best efforts, using RNA-seq and pathway analysis of differentially expressed genes, we were unable to conclusively identify candidate mechanisms that could explain why Resolved mice are able to dampen the acute inflammatory response so quickly. Nevertheless, there was some evidence of modest differential activation of anti-inflammatory pathways in Resolved mice at 24 hpi. We have added text in the Results and a paragraph in the Discussion section to address this point and discuss potential reasons why this question remains unanswered.

2) Figure 3: Some technical aspects on the transcriptomics are missing like the RNA quality (to be added in the Materials and methods) and a supplementary file giving the detail of all the libraries generated (with the barcode associated).

We have added details of the RNA quality to the Materials and methods and included details on the libraries in a new Supplementary file 3.

3) Figure 3: It would be interesting to compare the results of 'adult Naïve PBS', 'Sensitized PBS' and 'Resolved PBS' with a PCA analysis. Since these are the baseline experiments, one would like to understand how the initial conditions vary between the different group of mice.

We have included a new panel C in Figure 3 and added text to the Results and figure legend accordingly. As we discuss in the new Figure 3—figure supplement 1, the results are very similar to the previous study by O’Brien et al. (2016).

4) Figure 3: Similar experiments have been reported by the authors previously – in the supplementary materials, is it possible to compare the present results with those published before?

We assume the reviewers mean the O’Brien et al. (2016) paper that we mention above. We have included a Venn diagram in the supplementary information showing the overlap between the two studies of both the differentially expressed genes (panel A) and the enriched pathways (panel B) in Figure 3—figure supplement 1. We have also added text to the Results section describing these data.

5) Figure 3: please integrate a PCA that compiles all the experiments to have a global view of the data.

We have added a PCA plot of all experiments to Figure 3D, and modified the text and figure legend accordingly.

6) Please include a model in the discussion detailing the disease progression for each group (Resolving vs. Sensitized).

Thank you for this suggestion. We have added a model figure to the discussion as Figure 8.